# Platelet Function Test Use for Patients with Coronary Artery Disease in the Early 2020s

**DOI:** 10.3390/jcm9010194

**Published:** 2020-01-10

**Authors:** Pierre Fontana, Marco Roffi, Jean-Luc Reny

**Affiliations:** 1Geneva Platelet Group, Faculty of Medicine, University of Geneva, 1206 Geneva, Switzerland; jean-luc.reny@hcuge.ch; 2Division of Angiology and Haemostasis, Geneva University Hospitals, 1205 Geneva, Switzerland; 3Division of Cardiology, Geneva University Hospitals, 1205 Geneva, Switzerland; marco.roffi@hcuge.ch; 4Division of General Internal Medicine, Geneva University Hospitals, 1205 Geneva, Switzerland

**Keywords:** platelet function test, precision medicine, antiplatelet drug, cardiovascular patient, acute coronary syndrome

## Abstract

In the field of antithrombotics, precision medicine is of particular interest, as it may lower the incidence of potentially life-threatening side effects. Indeed, antiplatelet drugs such as P2Y_12_ inhibitors are one of the most common causes of emergency admissions for drug-related adverse events. The last ten years have seen a continuous debate on whether platelet function tests (PFTs) should be used to tailor antiplatelet drugs to cardiovascular patients. Large-scale randomized studies investigating the escalation of antiplatelet therapies according to the results of PFTs were mostly negative. Potent P2Y_12_ inhibitors are recommended as a first-line treatment in acute coronary syndrome patients, bringing the bleeding risk at the forefront. De-escalation from prasugrel or ticagrelor to clopidogrel is now considered, with or without the use of a PFT. This review covers recent advances in escalation and de-escalation strategies based on PFTs in various clinical settings. It also describes the main features of the most popular platelet function tests as well as the potential added value of genetic testing. Finally, we detail practical suggestions on how PFTs could be used in clinical practice.

## 1. Introduction

Precision (or personalized) medicine has been gaining ground in recent years thanks to European [1] and American [2] initiatives; it refers to a medical model using the characterization of an individual’s phenotype and/or genotype to tailor a therapeutic strategy [1,2,3]. In the field of antithrombotics, precision medicine is of particular interest [3] as it may lower the incidence of potentially life-threatening side effects. Indeed, bleeding events related to antiplatelet drugs such as P2Y_12_ inhibitors are one of the most common causes of emergency admissions for drug-related adverse events [4].

Dual antiplatelet therapy (DAPT), combining aspirin and a P2Y_12_ inhibitor, is the cornerstone in the prevention of ischemic events after an acute coronary syndrome (ACS) and/or a percutaneous coronary intervention (PCI). Depending on the clinical setting, the duration of DAPT may range from one month to more than one year [5,6]. Data from major trials and registries have shown that approximately 9%–10% of patients receiving DAPT experienced a thrombotic event within one year, whereas bleeding events occurred in about 2% of patients [3,7,8]. This suggests that individualized antiplatelet regimens, tailoring both DAPT potency as well as duration, might be beneficial in terms of net clinical benefit (i.e., the combination of ischemic and bleeding events). Several scores have been developed to identify patients at risk of bleeding or recurrent thrombotic events and who would benefit from a personalized approach to anti-P2Y_12_ selection and/or duration of DAPT. These scores have several drawbacks, however. First, their predictive value is, at best, moderate, with C-statistic values ranging from 0.6 to 0.7, depending on the validated cohort used [3,9,10,11]. Second, bleeding and thrombosis risk-scores share several common risk factors which make assessing the balance between ischemic and bleeding risks challenging for clinicians. Third, the clinical impact of these risk-prediction models has never been assessed as part of a clinical decision-making strategy in a prospective randomized controlled trial (RCT) [3,12].

The use of platelet function assays to tailor antiplatelet drugs to cardiovascular patients has been debated for the last ten years. Large-scale randomized studies investigating the escalation of antiplatelet therapy according to the results of platelet function tests (PFTs) mitigated the potential clinical usefulness of these assays. Indeed, the 2011 American College of Cardiology/American Heart Association guidelines released a Class IIb recommendation for the use of PFTs among selected patients taking P2Y_12_ inhibitors [13], but this was downgraded to a Class III recommendation in 2016 [5]. In cases of ACS, the latest European guidelines suggest that de-escalation, but not escalation, of P2Y_12_ inhibitors guided by a PFT may be considered, with a class IIb grading [6].

This review will briefly address milestones in the history of the PFT debate regarding cardiovascular patients and present our view on how PFTs should be used in the early 2020s, focusing on post-PCI management.

## 2. The Early Evidence for Platelet Function Testing

Three decades ago, one of the first reports suggesting variability in the platelet function of cardiovascular patients described this issue using 82 patients being treated with aspirin following a stroke [14]. Two years later, the same authors published the results of a prospective follow-up study of 180 stroke patients, 11% of whom had high platelet reactivity (HPR) at baseline despite aspirin treatment at a dose of 3 × 500 mg/day. During the two-year follow-up, major ischemic endpoints occurred in 4% of the patients without HPR and 40% of patients with HPR [15]. This landmark study kicked off of intensive investigation of this phenomenon, which was termed *aspirin resistance*. It was followed by another case series of 326 cardiovascular patients taking aspirin (325 mg/day) among which 17 (5.2%) were deemed *aspirin resistant* according to PFT. After a mean follow-up of 1.9 years, these patients showed a significantly higher rate of death, myocardial infarction, or stroke compared with non-HPR individuals (hazard ratio (HR) = 4.14, 95% CI 1.42–12.06, *p* = 0.009) [16].

These results in aspirin-treated patients prompted the investigation of the second most prescribed antiplatelet drug: clopidogrel. The first study addressed this issue using various PFT in 96 aspirin-treated patients receiving a loading dose of clopidogrel (300 mg) followed by a regular maintenance dose (75 mg/day) before elective coronary artery stenting [17]. The authors showed that around 30% of patients had HPR, according to their definition. Interestingly, they also showed that patients with the highest pre-treatment platelet reactivity remained the most reactive at 24h after clopidogrel treatment [17]. This raised the issue of whether environmentally and/or genetically related factors were associated with the variability in platelet reactivity, even without antiplatelet drugs. Indeed, this variability had already been described in the Framingham study [18], and several other association studies linked platelet receptor gene polymorphisms with platelet function [19,20,21].

The variability of on-treatment platelet reactivity has been particularly well investigated among patients on clopidogrel and is reviewed in [22]. Several studies were performed to identify the determinants of an HPR phenotype. Overall, HPR in patients treated with clopidogrel is only partially explained by individual factors such as body-weight or a high body mass index, diabetes, renal failure, increased C-reactive protein, old age, low left-ventricular ejection fraction, or a recent myocardial infarction [22,23,24]. Together, these individual factors explain only 10% of the total variance in platelet response to clopidogrel [22]. Genetic variants of the P2Y_12_ receptor gene were a priori good candidates to explain clopidogrel’s pharmacodynamic variability. The P2Y_12_ receptor’s first genetic variants were described in 2003 and were associated with platelet aggregation induced by adenosine diphosphate (ADP) [20] and peripheral arterial disease [25]. However, subsequent studies showed that the receptors genetic polymorphisms explained little, if any, of the variability in response to clopidogrel [22,26,27]. Since clopidogrel is a prodrug requiring bioactivation via several liver cytochromes (CYPs), genetic investigations scrutinized genes coding for those CYPs which led to the discovery of a *CYP2C19* genetic variant (*CYP2C19*2*, rs4244285) associated with the biological response to clopidogrel, first described in 2006 [28] and extensively studied since then [22]. Overall, this variant’s impact seems of limited importance. Indeed, in one genome-wide association study, the *CYP2C19*2* genotype accounted for 12% of the variation in response to clopidogrel, similar to the contribution of factors such as age, body mass index, and lipid levels [22,29]. In another study, of 534 stable cardiovascular patients treated with clopidogrel, the *CYP2C19*2* genotype accounted for 5% of the variation in response to that drug [22,30]. These data were further supported by a study of 760 cardiovascular patients treated with clopidogrel; it found that the *CYP2C19*2* genotype accounted for only 5.2% of the platelet reactivity phenotype [31]. The genetic background of this variability is indeed probably multigenic, as shown by a recent collaborative study [32].

## 3. A Broad Spectrum of Platelet Function Tests

There are several PFTs available for evaluating the biological effects of antiplatelet drugs (see Table 1) [33]. Their differences are mainly related to the different facets of platelet function that they explore; therefore, PFTs are not interchangeable. They can be classified as *target-centered* when focused on the targeted effects of aspirin or anti-P2Y_12_ drugs, or *non-target-centered*—mostly aggregation-based—when they are more integrative and evaluate the overall platelet aggregation process.

Currently, the most target-centered assay for assessing aspirin’s biological response relies on the quantification of serum TxB_2_, a stable metabolite of TxA_2_ obtained after incubation of whole blood at 37 °C for 1 h. Using this assay, aspirin’s inability to suppress TxA_2_ generation is very rare [34,35]. However, several clinical situations are associated with this biological phenomenon, such as non-compliance, the use of preparations with enteric coatings, drug–drug interactions with nonsteroidal anti-inflammatory drugs (NSAIDs), or increased platelet turnover [36]. Diabetes may also be associated with impaired aspirin efficacy through glycation of the COX-1 enzyme that hinders aspirin reaching the protein’s Ser529 residue and thus the acetylation process [37]. Despite the overall homogenous inhibition of platelet-derived TxA_2_ by aspirin, non-target-centered PFTs are of interest since they show a more variable inter-individual phenotype. These methods may thus capture non-TxA_2_-dependent platelet activation pathways mediating platelet aggregation despite adequate inhibition of TxA_2_ by aspirin [34,38]. For example, the PFA-100^®^ (Siemens, Germany) has been particularly well investigated since it is an easy-to-use point-of-care assay (see Table 1). According to a meta-analysis of 15 studies using this PFT, approximately 30% of aspirin-treated patients showed normal results (i.e., *aspirin resistant*) [39]. As expected, residual concentrations of TxB_2_ did not correlate with PFA-100^®^ results [34].

Since clopidogrel, prasugrel, and ticagrelor target the ADP P2Y_12_ receptor, PFTs evaluating the biological effects of these drugs use ADP as the agonist. The vasodilator-stimulated phosphoprotein (VASP) assay is the most target-centered assay evaluating the degree of P2Y_12_ receptor inhibition [40]. Other ADP-induced PFTs have a poor, or, at best, a moderate, correlation with this reference test [41]. Similarly to the PFTs used for patients treated with aspirin, they are not interchangeable. The prevalence of HPR in patients treated with clopidogrel ranged from 16%–50% according to the PFT used, the cut-off value, and various—mostly unidentified—additional factors [42].

Prasugrel is a third-generation thienopyridine. It is more potent than clopidogrel, but its pharmacodynamics’ variability is equally high. This has been best demonstrated in pharmacokinetic/pharmacodynamic studies [22,43] showing similar standard deviations for maximal ADP-induced aggregation—and with the VASP assay—among patients treated with clopidogrel and prasugrel [43]. The variability in response to ticagrelor is still unclear, but it seems to be weaker than for clopidogrel and prasugrel [22,44]. Although ticagrelor does not require bio-activation as thienopyridines, one of its metabolite (ARC24910XX) is mediated via CYP3A4 and is also a potent P2Y_12_ inhibitor [22]. Since prasugrel and ticagrelor are more potent than clopidogrel, the prevalence of HPR with these agents is lower. In a meta-analysis of almost 2000 patients, only 1.5% and 9.8% of patients treated with ticagrelor and prasugrel had HPR, respectively, as measured using the VASP and the VerifyNow^®^ assays [45]. The concept of low on-treatment platelet reactivity (LPR) thus emerged logically and defined patients with lower than expected on-treatment platelet reactivity; this can reach up to 70% of patients treated with prasugrel or ticagrelor when measured using impedance aggregometry [46].

As PFTs explore different platelet activation pathways and do not correlate well with one another, the prevalence of HPR and LPR varies considerably depending on the test. Pragmatically, all interventional studies aiming to tailor antiplatelet drug treatments with reference to platelet reactivity measurements were performed using easy-to-use, standardized, point-of-care assays using consensus cut-offs that had been chosen by investigators [47].

## 4. Platelet Function Tests: Clinical Association Studies

### 4.1. PFTs and Ischemic Events

Some observational studies have shown associations between HPR in patients treated with aspirin and the occurrence of ischemic events. These studies were summarized in a meta-analysis which showed nearly a four-fold increase in the risk of a recurrence of ischemic events among those patients deemed to be aspirin resistant [48]. However, as previously mentioned, the definition of aspirin resistance is heterogeneous across the various PFTs used, as demonstrated by the high heterogeneity index of the study results compiled in the meta-analysis [48]. When only one PFT is considered at a time, this heterogeneity disappears, as revealed by the smaller confidence interval (CI) around the odds ratio (OR) [39]. The association between platelet reactivity and cardiovascular events has been less well explored using aspirin-dedicated PFTs than using anti-P2Y_12_-dedicated PFTs. Indeed, studies of the association between the levels of serum TxB_2_, a target-centered marker of the platelet response to aspirin, and the occurrence of ischemic events are scarce and conflicting [49,50]. Nevertheless, a recent meta-analysis involving 11 studies and 11,857 coronary patients treated with aspirin showed that aspirin resistance was associated with an increased risk of all-cause death (OR = 2.42, 95% CI 1.86–3.15) and target vessel revascularization (OR = 2.20, 95% CI 1.19–4.08) [51]. However, it seems that the level of baseline cardiovascular risk is important for the clinical relevance of HPR in patients treated with aspirin [52], a concept that has been investigated in depth among patients treated with clopidogrel.

Indeed, there are numerous studies and meta-analyses [53,54,55] on the associations between HPR and cardiovascular outcomes among patients treated with P2Y_12_ inhibitors. As often in clinical research, the first observational studies revealed the high relative risks or ORs of HPR for the outcome of ischemic events. This was consistent across a wide array of both target-centered and non-target-centered PFTs. Interestingly, these relative risks became lower as more recent and larger association studies were conducted, and this time-effect was significant in a meta-regression [53]. Importantly, and similarly to what has been found among patients treated with aspirin [52], HPR was not prognostic of recurrent ischemic events in stable cardiovascular patients treated with clopidogrel [50]. This led to the hypothesis that the clinical setting and the level of baseline cardiovascular risk may be critical to the clinical relevance of HPR and the selection of patients for interventional studies [56]. A meta-analysis of individual data from more than 6000 patients [57] showed that HPR was not prognostic of recurrent ischemic events in low-risk cardiovascular patients treated with clopidogrel. While the low-risk patients had a 2% yearly risk of recurrent ischemic events, regardless of the presence of HPR, the prognostic value of HPR increased in a dose-dependent fashion, as the number of risk factors and level of baseline cardiovascular risk increased. In higher-risk patients, with two or more risk factors (including age > 75, hypertension, diabetes, and ACS), the yearly incidence of ischemic events rose from 2.5% to 8% in those with lower and higher PR, respectively. The presence of ACS at inclusion was associated with the higher hazard ratio of recurrent ischemic events. This clinical presentation may be a major contributor to the interplay between HPR and the clinical outcome [57].

### 4.2. PFTs and Bleeding Events

Clinical association studies have linked LPR (as established using aspirin-dedicated PFTs) with bleeding; these used aggregation-based assays and were conflicting. Two large registries, the ADAPT-DES and ISAR-ASPI studies, found a borderline-significant increased risk associated with LPR in the former (hazard ratio = 0.65, 95% CI 0.43–0.99) [58] but not in the latter [59]. Regarding P2Y_12_ inhibitors, and as mentioned above, LPR may be quite frequent with next-generation drugs prasugrel and ticagrelor, but the association with bleeding events has been studied more among patients treated with clopidogrel. The available evidence is more consistent than when using PFTs targeting aspirin response, and the largest meta-analysis, gathering individual data from more than 20,000 cardiovascular patients, clearly showed that LPR on clopidogrel was associated with an increased risk of bleeding events (risk ratio = 1.74, 95% CI 1.47–2.06) [55]. All this work has led to the concept of a platelet reactivity *sweet spot* where the net clinical benefit of treatment is highest, while ischemic and bleeding events occur more frequently in HPR and LPR patients, respectively [47].

## 5. Do Platelet Function Tests Help to Avoid Ischemic Events?

These clinical association studies raised the hypothesis that tailoring antiplatelet therapy according to PFT results might decrease the risk/benefit ratio. Early RCTs involving personalized clopidogrel dosing following a PFT were performed among PCI patients, half of whom were suffering from an ACS at inclusion [60,61]. The personalized intervention in these trials led to a total 2400 mg clopidogrel loading dose over a few days before PCI in some patients, an unprecedented and never renewed dose. This intervention showed a 10% absolute reduction of recurrent ischemic events at 30 days, with no increase in bleeding. This original evidence led to much larger trials involving milder dose adjustments after PCI (rather than before it) among patients who were at lower cardiovascular risk [62,63,64]. These latter RCTs showed no benefits to the use of personalized antiplatelet regimen based on PFTs. A meta-analysis of 13 RCTs and more than 7000 patients showed that a personalized, intensified antiplatelet regimen, based on a PFT, decreased the incidence of cardiovascular events without increasing the risk of bleeding [65]. Potential biases in some trials, together with some statistical heterogeneity, may mitigate this interpretation. More importantly, incidences of cardiovascular events across these different trials were highly heterogeneous, possibly related to differences in the cardiovascular risk levels and lengths of follow-up. Altogether, the evidence is currently too weak to support the routine use of PFTs for deciding on personalized anti-P2Y_12_ drug treatment for all cardiovascular patients, particularly those without a high cardiovascular risk (such as those without ACS), for whom clopidogrel is recommended [66]. Moreover, most of the trials used clopidogrel as the first line P2Y_12_ inhibitor, even among patients with ACS. Since in ACS patients, prasugrel or ticagrelor are preferred over clopidogrel and recommended as the first line anti-P2Y_12_ treatment, any concern regarding HPR in these patients is mitigated.

Among chronic coronary syndrome patients undergoing stenting, dual antiplatelet therapy with aspirin and clopidogrel remains the standard of care (Class IA grading), whereas more potent P2Y_12_ inhibitors are only suggested as options for patients with specific high-thrombotic risk or for selected complex stenting procedures with a low level of evidence (Class IIbC) [66].

## 6. Do Platelet Function Tests Help to Avoid Bleeding Events?

In recent years, evolving stent technology has led to a dramatic reduction in stent thrombosis rates and other stent-related ischemic events [67]. In the current era of potent P2Y_12_ inhibitors, bleeding risk is now the most frequent adverse event among patients with ACS after stenting. This has led to the emergence of alternate antithrombotic strategies, such as *de-escalation*. De-escalation aims to minimize exposure to DAPT involving potent P2Y_12_ inhibitors (prasugrel or ticagrelor) by replacing them with clopidogrel in selected clinical settings where bleeding risk is a particular concern and outweighs the thrombotic risk. The concept of de-escalation has been supported by pivotal trials such as the TRITON-TIMI 38 study, which showed that the benefit of prasugrel over clopidogrel in terms of ischemic event reduction was mainly observed within the first 30 days, while the associated increased risk of bleeding persisted over the follow-up period [68]. In the PLATO trial, the reduction in ischemic risk was more homogenous during the follow-up, and bleeding events on ticagrelor occurred predominantly during the maintenance phase [69].

These results, together with findings that variability in platelet reactivity did not predict ischemic events in stable cardiovascular patients [50,52,70,71], raised the possibility of reducing the intensity of antiplatelet therapy after the acute phase. The TOPIC study included 646 ACS patients treated with aspirin and ticagrelor or prasugrel one month after PCI [72]. Patients were randomized either to continue DAPT with their current P2Y_12_ inhibitors or to switch to a DAPT with clopidogrel. After one year of follow-up, the primary composite endpoint of ischemic and bleeding events occurred less frequently among patients assigned to the switched DAPT (13.4%) than among the control group (26.3%, *p* < 0.01) [72]. This effect was mostly driven by the 60% decrease in bleeding events in the switched group, without an increase in ischemic events. It is of note that the switched DAPT strategy was superior in terms of bleeding regardless of the initial platelet reactivity [73]. The recent TWILIGHT study further supported a strategy based on a shorter DAPT [74]. This study randomized more than 7000 high-risk patients who had undergone an uneventful 3-month course of DAPT after PCI using ticagrelor, to either continue with DAPT for an additional 9 months or stop aspirin. The primary endpoint (bleeding events) occurred less frequently in the monotherapy arm than in the DAPT arm (hazard ratio = 0.56, 95% CI 0.45–0.68, *p* < 0.001), with no difference in ischemic endpoints (*p* < 0.001 for non-inferiority). This strategy of short, uniform de-escalation, irrespective of PFT results, has also been supported by other recent trials involving several thousand patients and showing that rapid de-escalation of antiplatelet therapy was effective and associated with fewer bleeding events after ACS, including STEMI [75,76].

TROPICAL-ACS was a landmark study for strategies involving de-escalation guided by PFTs; its 2610 ACS patients were randomized to a conventional DAPT arm with prasugrel or a strategy guided by a PFT. In the latter arm, 1306 patients began DAPT with prasugrel for 7 days and then switched to clopidogrel for a further 7 days. A PFT was then performed, and patients with HPR (*n* = 511, 39%) were switched back to prasugrel. The primary endpoint was a composite of cardiovascular death, myocardial infarction, stroke, or bleeding events throughout the 1-year follow-up. The study showed that the strategy guided by PFT was not inferior to standard DAPT, with the primary endpoint occurring in 7% of the de-escalation group and 9% of the control group. There were no significant differences between the arms regarding either ischemic or (more surprisingly) bleeding events. This study suggested that a de-escalation strategy guided by a PFT was an option for patients with ACS with estimated comparable ischemic and bleeding risks if treated for 12 months with a potent P2Y_12_ inhibitor such as prasugrel.

It is noteworthy that none of these studies was powered to assess safety in terms of a recurrence of an ischemic event, which is the more feared complication when dealing with a de-escalation strategy. Nevertheless, the overall rate of ischemic events was low, mitigating any concerns regarding the thrombotic risk associated with de-escalation, with or without PFT.

## 7. Genetic Tests or Platelet Function Tests?

Genetic testing aims to predict the pharmacodynamic effects of a given antiplatelet drug without the need for a PFT. It has several key advantages over PFTs, including quickly available, unequivocal results using current point-of-care assays, the absence of inter-individual variability, and, most importantly, there being no need to potentially challenge the patient with an antiplatelet drug for several days before a PFT. However, and as mentioned above, the currently available knowledge of the genetic background governing antiplatelet drug pharmacodynamics only explains between 3.5% and 12% of clopidogrel response [29,30,31,32], and the present genetic panel available poorly reflects the drug’s pharmacodynamics. Nevertheless, several trials have investigated the clinical impact of a strategy based on genetics to tailor antiplatelet drug treatment in de-escalation settings. In one of the largest studies, POPULAR-GENETICS, 2488 STEMI patients were randomized within 48 h of revascularization either to a group receiving standard DAPT including prasugrel or ticagrelor after PCI or to a group guided by genotype where *CYP2C19*2* carriers were continued on either prasugrel or ticagrelor while non-carriers (67% of the patients) were prescribed clopidogrel instead. All patients received ticagrelor or prasugrel at the time of PCI, drugs that were continued until randomization. The study showed that the approach guided by genotype was not inferior to the standard approach in terms of net clinical benefits (a composite of ischemic and bleeding events), and it displayed fewer (mostly minor) bleeding events (9.8% vs. 12.5%; HR = 0.78, 95% CI 0.61–0.98, *p* = 0.04). Another large study of a similar design—the PHARMACLO trial—included patients with ACS, but the intervention arm included screening for three genetic polymorphisms of the *2C19* and *ABCB1* genes, as well as consideration of clinical parameters with which the treating physician would select the P2Y_12_ inhibitor after PCI. This study showed promising results, with a composite of ischemic and bleeding outcomes occurring in 15.9% of patients allocated to genetic testing and in 25.9% of those receiving standard care (HR = 0.58, 95% CI 0.43–0.78, *p* < 0.001). It is of note that after enrolling only 888 patients (25% of the planned population), this study was terminated prematurely because of a legal issue regarding the genetic assay. In addition, clopidogrel was often prescribed, including more frequently in the standard-care arm (50.7% vs. 43.3%), which did not comply with current guidelines.

Finally, a sub-study of the TROPICAL-ACS study observed no added benefits from using genotyping to predict ischemic and bleeding risks among patients who had undergone a de-escalation treatment guided by a PFT [77]. This latter finding was consistent with the fact that platelet reactivity is determinant for cardiovascular outcome, and that genotyping only poorly reflects the results of PFTs. Indeed, genetic testing is associated with poor predictions of HPR, with a sensitivity of < 40% yielding a negative predictive value of about 50% [78]. In head-to-head comparisons with genetic testing, PFTs emerge as superior, albeit imperfect, predictors of clinical outcome [79]. Therefore, until more robust data arises from ongoing trials, we currently recommend that tailoring an antiplatelet strategy need only involve a PFT [80]. Future directions aiming to identify the role of genetic testing in this field include the analysis of gene–gene interactions at a biological pathway level rather on single individual variants as well as gene–environmental interactions [81].

## 8. Implementation of Platelet Function Testing in Clinical Practice

The current recommended first-line treatment combining the low thrombogenicity of the latest stents with potent P2Y_12_ inhibitors has almost closed the discussion about antithrombotic drug escalation based on PFTs. For most patients, the main challenge and the foremost risk is now bleeding. PFTs may be useful for deciding on a de-escalation from prasugrel or ticagrelor to clopidogrel. However, trials investigating de-escalation strategies have so far lacked the power to assess PFTs’ impact on thrombotic events. Therefore, de-escalation strategies should be restricted to patients with a non-low bleeding risk.

In daily practice, considering that ACS is a major factor in cardiovascular risk-level assessment, that next-generation P2Y_12_ inhibitors are not licensed for PCIs without ACS, and that interventional studies have not shown that low-risk patients get any benefits from a tailored treatment approach based on PFTs, we recommend that PFTs not be performed on patients without ACS (see Figure 1). For patients with ACS and at a low risk of bleeding, DAPT with prasugrel or ticagrelor is recommended without recourse to a PFT. For patients with ACS and a non-low risk of bleeding, several options are suggested (see Figure 1). For patients with a non-low risk of bleeding and in equipoise regarding the net clinical benefits of a prolonged DAPT with prasugrel or ticagrelor (e.g., a patient with a history of gastric ulcers with no active bleeding when treated with a proton pump inhibitor), a PFT should be considered and next-generation P2Y_12_ inhibitors prescribed only if the patient is deemed a poor responder to clopidogrel (TROPICAL-ACS study management). For patients with the highest bleeding/thrombotic risk ratio (for example a patients with colonic angiodysplasia with previous history of several bleeding events), DAPT with clopidogrel seems the best option, without a PFT, since escalating antiplatelet therapy, in this case, would likely be associated with a major increase in the risk of bleeding. For other patients, a uniform, short-term DAPT with prasugrel or ticagrelor, followed by de-escalation, should be considered (TOPIC or TWILIGHT study management).

## Figures and Tables

**Figure 1 jcm-09-00194-f001:**
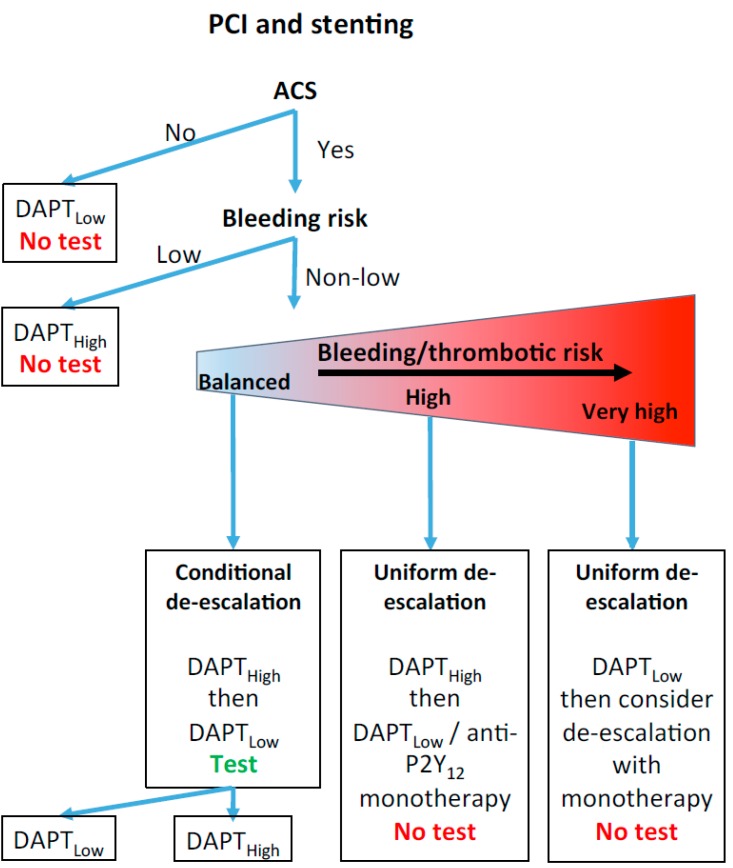
Antiplatelet drug strategy and the role of platelet function tests after a percutaneous coronary intervention. See main text for details. ACS: acute coronary syndrome. DAPT_High_: dual antiplatelet therapy with aspirin and prasugrel or ticagrelor. DAPT_Low_: dual antiplatelet therapy with aspirin and clopidogrel. PCI: percutaneous coronary intervention.

**Table 1 jcm-09-00194-t001:** Succinct description of the main platelet function tests used for the in vitro evaluation of the effects of antiplatelet therapy; adapted from [33], with permission.

Platelet Function Test	Principle and Interpretation
Conventional photometric aggregation	Changes in light transmission in platelet-rich plasma. Activators: arachidonic acid for aspirin, ADP for P2Y_12_ inhibitors, or another activator (collagen, TRAP) that more or less uses activation amplification (thromboxane A_2_ and ADP).
Serum thromboxane B_2_	Coagulation of whole blood at 37 ℃ and measurement of thromboxane B_2_ (a stable metabolite of thromboxane A_2_) in the serum obtained. Close evaluation of the aspirin target (COX-1), but it can also be diminished by an NSAID other than aspirin or poor blood coagulation.
VASP assay	ADP-induced inhibition, via its interaction with the P2Y_12_ receptor, of the elevation of intra-platelet levels of cAMP (a secondary messenger inhibiting platelet activation) induced by PGE1 (platelet activation inhibitor); then, detection by quantification of the degree of phosphorylation of the VASP protein, using flow cytometry or ELISA. A whole blood test that specifically evaluates the ADP P2Y_12_ receptor’s inhibitors.
VerifyNow^®^	Automated measurement, in whole blood, of the consequence of the interaction between fibrinogen and activated GP IIb/IIIa complex (artificial microbeads covered with fibrinogen). Dedicated cartridges for treatment using aspirin, P2Y_12_, or anti-GPIIb/IIIa inhibitors.
Impedance aggregometry	Whole blood (diluted), two devices: Multiplate^®^ and ROTEM^®^ platelet.Multiplate^®^ is a multiple electrode impedance platelet aggregometer with five channels and computer-assisted control. Platelet activators to evaluate the effects of aspirin or of P2Y_12_ inhibitors (arachidonic acid and ADP, respectively). ROTEM^®^ platelet is a new module of the ROTEM^®^ device dedicated to platelet function.
PFA^®^	PFA-100^®^ (now PFA-200^®^); whole blood under flow, with (very) high shear stress. A platelet plug occludes an orifice in a membrane soaked with either collagen and ADP or collagen and epinephrine. Sensitive to aspirin when using a collagen and epinephrine cartridge, but not very sensitive to P2Y_12_ inhibitors; there is a sensitized cartridge dedicated to P2Y_12_ inhibitors (INNOVANCE^®^ PFA P2Y).
TEG^®^ PlateletMapping™	Gradual modification of the viscoelastic properties of whole blood, along with its coagulation and clot organization (its mechanical properties). Sensitized evaluation of platelet involvement in maximal amplitude.

Some tests that have only been used in particular studies and are not widely available have not been included in this table. It is important to note that activator concentrations can differ from one test to another. ADP: adenosine diphosphate; cAMP: cyclic adenosine monophosphate; COX: cyclooxygenase; ELISA: enzyme-linked immunosorbent assay; GP: glycoprotein; NSAID: non-steroidal anti-inflammatory agent; PFA: Platelet Function Analyzer; PGE: prostaglandin E; ROTEM: rotational elastometry; TEG: Thromboelastograph; TRAP: thrombin receptor-activating peptide. VASP: vasodilator-stimulated phosphoprotein.

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
