# Peer review of "Platelet Function Test Use for Patients with Coronary Artery Disease in the Early 2020s"

_jcm, 2020, doi:10.3390/jcm9010194_

Round 1
Reviewer 1 Report
Fontana et al. present a narrative review of platelet function testing in the context of personalized antiplatelet therapy. The manuscript reads well and the authors cover essential literature on this interesting topic.
Please find specific minor suggestions below:
1 – Consider adding a timeline figure for ASA and P2Y12 inhibitors separately outlining major practice-changing concepts / papers.
2 – On p. 3, last paragraph. I assume “showed normal results” translates to were aspirin resistant?
3 – On p. 4, ticagrelor does have interactions with CYP3A-inducing agents. See Pourdjabbar et al. PMID: 27088404.
4 – On p. 5, the meta-analysis used to support association of PFTs with outcomes in patients on aspirin does not include major studies by Frelinger et al. (n=700, PMID 19996015) and Reny et al. (n=771, PMID 22615340). Perhaps worth addressing separately. The work of Dretzke et al (PMID: 25984731) might also offer interesting insights.
5 – On p. 7, there might be gene-gene interactions that are not captured by candidate gene testing. Attempts at whole exome have however been disappointing so far. Perhaps worth referring to NHLBI position paper (PMID: 23008471) on future directions for research in this field?
Author Response
Responses to Reviewer 1
We thank Reviewer 1 for her/his helpful comments and we provide a point-by-point response below.
1 – Consider adding a timeline figure for ASA and P2Y12 inhibitors separately outlining major practice-changing concepts / papers.
We have considered adding this figure but we fell that it would rather fit with a review on antiplatelet therapy rather than platelet function testing. Moreover, the most recent clinical trials addressed the issue of the duration, the role of pre-treatment before PCI and the type of P2Y12 inhibitors that is beyond the scope of the present review.
2 – On p. 3, last paragraph. I assume “showed normal results” translates to were aspirin resistant?
This is correct, we have clarified this issue in the revised manuscript.
3 – On p. 4, ticagrelor does have interactions with CYP3A-inducing agents. See Pourdjabbar et al. PMID: 27088404.
We fully agree with this comment and the corresponding sentence has been modified accordingly.
4 – On p. 5, the meta-analysis used to support association of PFTs with outcomes in patients on aspirin does not include major studies by Frelinger et al. (n=700, PMID 19996015) and Reny et al. (n=771, PMID 22615340). Perhaps worth addressing separately. The work of Dretzke et al (PMID: 25984731) might also offer interesting insights.
These two studies (PMID 19996015 and PMID 22615340) were indeed quoted in the previous sentence (lines 175-178 of page 5) and are discussed separately as conflicting.
5 – On p. 7, there might be gene-gene interactions that are not captured by candidate gene testing. Attempts at whole exome have however been disappointing so far. Perhaps worth referring to NHLBI position paper (PMID: 23008471) on future directions for research in this field?
This is a very insightful comment and we have elaborated on this issue in the revised manuscript page 8. Of note, we have a manuscript in revision that deals with this topic (project described here: http://p3.snf.ch/project-153206). The corresponding reference will be added if the manuscript is accepted before the edition of the present review.

Reviewer 2 Report
Platelets play an essential role in primary haemostasis. DAPT is indicated following ACS and after stent insertion to minimize the risk of trombosis incidence.
In this review the authors aimed to advances in escalation and de-escalation DAPT strategies based on PFTs based on various clinical settings. There is a lack of information about other clinical conditions, except PCI where PFTs are helpful in clinical decision-making for instant to support transfusion guidelines or perioperative settings. For instance, Ranucci et al. reported that investigating PF by using multiple electrode aggregometry before cardiac surgery has made it possible to identify hiper-responder patients with high risk of postoperative bleeding.
Minor issues:
In Table 1. authors enumerate different activators. One of them, TRAP test is sensitive to GPIIbIIIa inhibitors only, and in the absence for these drugs, represents the „natural” aggregation potential of the platelet. Many algorithms in PFTs involve AA or ADP tests concurrent with TRAP test to describe a broad spectrum of PF. Abbreviation TRAP in Table 1 should be also explained as thrombin receptor-activating peptide.
References:
Please describe more precisely citation number 3. Should be:… 2019,58,1517-1532, and probably without doi: 10.1007/s40262-019-00792-y.
Overall publication value:
Authors present well written paper based on revision of numerous studies and meta-analyses. Presented algorithm „Antiplatelet drug strategy of de-escalation” is easy to implement and should make PFTs strategy more cost-effective.
Author Response
Responses to Reviewer 2
In this review the authors aimed to advances in escalation and de-escalation DAPT strategies based on PFTs based on various clinical settings. There is a lack of information about other clinical conditions, except PCI where PFTs are helpful in clinical decision-making for instant to support transfusion guidelines or perioperative settings. For instance, Ranucci et al. reported that investigating PF by using multiple electrode aggregometry before cardiac surgery has made it possible to identify hiper-responder patients with high risk of postoperative bleeding.
We agree that PFT may be useful in cardiac surgery setting. However, and as stated in the abstract and introduction sections, we addressed more specifically the role of PFT in the escalation and de-escalation of antiplatelet therapy. We have added a sentence at the end of the introduction that clarify this in the revised manuscript.
Minor issues:
In Table 1. authors enumerate different activators. One of them, TRAP test is sensitive to GPIIbIIIa inhibitors only, and in the absence for these drugs, represents the „natural” aggregation potential of the platelet. Many algorithms in PFTs involve AA or ADP tests concurrent with TRAP test to describe a broad spectrum of PF. Abbreviation TRAP in Table 1 should be also explained as thrombin receptor-activating peptide.
We have now added the explanation of the abbreviation.
References:
Please describe more precisely citation number 3. Should be:… 2019,58,1517-1532, and probably without doi: 10.1007/s40262-019-00792-y.
The reference has been corrected.
